# Reporting conflicts of interest in randomised trials of patient blood management interventions in patients requiring major surgery: a systematic review and meta-analysis

Marius Roman  ,[1] Oluwatomini Fashina,[1] Sara Tomassini,[1] Riccardo G Abbasciano,[1] Florence Lai  ,[1] Toby Richards,[2] Gavin Murphy[1]

[1]Department of Cardiovascular Sciences and NIHR Leicester Biomedical Research Unit in Cardiovascular Medicine, University of Leicester, Leicester, UK
[2]Department of Vascular Surgery, The University of Western Australia, Perth, Western Australia, Australia

**Correspondence to**
Dr Marius Roman;
mr345@le.ac.uk

## ABSTRACT

**Objective** This study aimed to systematically review the effects of declared and undeclared conflicts of interest on randomised controlled trials (RCTs) of patient blood management (PBM) interventions.

**Design** We performed a secondary analysis of a recently published meta-analysis of RCTs evaluating five common PBM interventions in patients undergoing major surgery.

**Data sources** The databases searched by the original systematic reviews were searched using subject headings and Medical Subject Headings terms according to search strategies from the final search time-points until 1 June 2019.

**Eligibility criteria** RCTs on PBM irrespective of blinding, language, date of publication and sample size were included. Abstracts and unpublished trials were excluded. Conflicts of interest were defined as sponsorship, funding or authorship by industry, professional PBM advocacy groups or blood services.

**Data extraction and synthesis** Three independent reviewers extracted the data and assessed the risk of bias. Pooled treatment effect estimates were reported as risk ratios (RRs) or standardised mean difference with 95% CIs. Heterogeneity was quantified using the I² statistic.

**Results** Three hundred and eighty-nine RCTs totalling 53 635 participants were included. Thirty-two trials (8%) were considered free from important sources of bias. There was reporting bias favouring PBM interventions on transfusion across all analyses. In trials with no declared author conflicts of interest, the treatment effect on mortality was RR 1.12 (0.86 to 1.45). In trials where author conflicts of interest were declared, the treatment effect on mortality was RR 0.84 (0.69 to 1.03), with significant reporting bias favouring PBM interventions. Trials with declared conflicts linked to professional PBM advocacy groups (five studies, n=977 patients) reported statistically significant reductions in mortality RR 0.40 (0.17 to 0.92), unlike other groups.

**Conclusions** Low certainty of the evidence that guides PBM implementation is confounded by evidence of reporting bias, and the effects of declared and undeclared conflicts of interest, favouring PBM on important trial outcomes.

## STRENGTHS AND LIMITATIONS OF THIS STUDY

⇒ This is the most comprehensive review to date of patient blood management (PBM) randomised controlled trials using Cochrane methodology showing reporting bias in favour of PBM interventions on transfusion and significant treatment effects on mortality where authors declared conflicts of interest.

⇒ Despite multiple settings and interventions, there was very little heterogeneity in the PBM impact on clinical outcomes.

⇒ The limitations include the low methodological quality of many of the studies, although similar treatment effects were observed when the analysis was restricted to groups at low risk of important bias.

⇒ This study relied on reported conflicts of interest in published trial reports for this analysis, and despite subgroup analyses and attempts to adjust for undeclared conflicts, these may have altered our results.

## INTRODUCTION

Patient blood management (PBM) describes the application of personalised, evidence-based, care bundles of interventions, aimed to optimise haemoglobin levels, reduce bleeding and transfusion with the specific intention of improving patient outcomes.[1 2] PBM is a patient-centred, systematic, evidence-based approach to improve patient outcomes by managing and preserving a patient's own blood, while promoting patient safety and empowerment. PBM has now become an established standard of care for blood transfusion practice in surgical patients.[2] However, randomised controlled trials (RCTs) comparing individual interventions as part of PBM interventions do not appear to demonstrate patient benefits beyond reductions in red cell transfusion.[2 3] Conflict of interest (COI) is defined as professional judgement

concerning a primary interest (such as patients' welfare or the validity of research) being influenced by a secondary interest (such as financial gain).[4] Perceptions of COI changed with the implementation of International Committee of Medical Journal Editors (ICMJE) guidelines on disclosure and reporting of COIs. Clinical trials with COIs may be subject to reporting biases or biased design due to the hypothesis, participants, interventions and outcomes tested.[5] Attempts to disseminate evidence of uncertainty are often challenged by advocacy groups and professional PBM bodies, which may raise the question of potential conflicts of interest, including those linked to professional PBM-related organisations or PBM-related healthcare consultancies.[6 7] We hypothesised that these conflicts may also influence the design, conduct and reporting of trials of PBM interventions in people requiring surgery. We tested this hypothesis in the dataset from a recently published comprehensive systematic review[3] and meta-analysis of trials of five common PBM interventions in people undergoing surgery. The aim of this study was to assess whether there may be reporting bias in RCTs of PBM intervention where the authors declare COI. We wished to assess the outcomes of RCTs in studies where there was perceived COI compared with those studies without apparent COI.

## METHODS

A systematic review of RCTs was performed using the methods described in Cochrane Handbook for Systematic Reviews of Interventions.[8] The review adhered to the Preferring Reporting Items for Systematic Reviews and Meta-Analyses guidelines.[9]

The following systematic reviews were updated :
- ► Cochrane review of iron therapy in patents without chronic kidney disease.[10]
- ► Cochrane review of restrictive red cell transfusion thresholds.[11]
- ► Cochrane review of cell salvage.[12]
- ► Systematic review of tranexamic acid in surgical patients.[13]
- ► Cochrane review of blood management algorithms based on point-of-care tests for coagulopathy.[14]
- ► The 2015 National Institute for Clinical and Healthcare Excellence (NICE, UK) Transfusion guideline review of studies evaluating the cost-effectiveness of PBM interventions.[15]

### Study eligibility

Studies were included if they fulfilled the inclusion criteria of a previous review conducted by our research group on PBM interventions in a population of patients undergoing major surgery.[3] Briefly, RCTs irrespective of blinding, language, publication status, date of publication and sample size investigating intervention targeting PBM interventions. PBM interventions were defined as: preoperative iron therapy, cell salvage and/or autotransfusion, restrictive transfusion thresholds, tranexamic acid and point-of-care testing for coagulopathy.

### Data sources

The following databases: Biosis, CENTRAL, CINAHL, ClinicalTrials.gov, Embase, LILACS, MEDLINE (OvidSP), Pubmed, Transfusion Evidence Library, Web of Knowledge, Web Of Science, WHO International Clinical Trials Registry Platform, ISRCTN Registry were searched using subject headings and Medical Subject Headings terms according to the original systematic reviews search strategies from the final search time-points until 1 June 2019. The full search strategy is detailed in the online supplemental appendix 1.

### Types of participants
#### Inclusion criteria

Patients of any age undergoing: cardiovascular, neoplastic, orthopaedic, gastrointestinal, urology, organ transplantation, plastic or maxillofacial surgery.

#### Exclusion criteria

Studies with patients undergoing treatment for trauma, burns or gastrointestinal haemorrhage, gynaecological/obstetrics procedures, dental procedures, or patients recruited from critical care, were excluded. Studies that used unwashed autologous red cells in trials of cell salvage, or comparing different tranexamic acid or iron formulations or doses without a control group were excluded. In studies comparing multiple formulations, the intravenous group was included if present, otherwise oral or other formulations were included. Studies that did not report the specified co-primary outcomes or that were not peer reviewed were excluded.

### Exposures of interest

All conflicts of interest were assessed by two independent assessors. Conflicts of interest were assessed based on the ICMJE standards for reporting conflicts of interest.

Conflict of Interest for Authorship was defined as employment, advisor/consultancy payments, speakers' fees, unspecified financial ties, honorariums, employee relationships, travel fees, stock ownership and patents. Conflict of Interest for Authorship for any author of each manuscript was determined from the study publication or a COI listed for the author in any other trial reported within 3 years of the study included in this review. COIs were categorised as: any, unclear or none declared.

Conflict of Interest for Funding was categorised as: any (Declared CONFLICT OF INTEREST related), none declared or unclear.

Conflict of Interest for Funding was determined from the published text or trial registry where available. Conflicts of Interest for Funding were further categorised as: industry, non profit (Academic Institution, Charity and Government), PBM advocacy groups, none stated or unclear. Studies partly funded by industry were classified as industry funded.

Patient Blood Management Advocacy Groups were categorised as: yes, no, unclear. Examples include the

Network for the Advancement of Transfusion Alternatives, the Society for the Advancement of Blood Management, the Society for Blood Management, World PBM Network, the Patient Blood Management Academy, (https://www.pbm-academy.de/en/), the National Anemia Action Council, Medical Society for Blood Management, Patient Blood Management European Network, International Foundation for Patient Blood Management (https://www.ifpbm.org/), Maturity Assessment Model in PBM (https://mapbm.org/public/home/en) and the Western Australia Patient Blood Management Group. PBM professional advocacy groups are composed of stakeholders with an interest in advancing and promoting alternatives to blood transfusion and PBM. In most cases, it is unclear how these organisations are funded or whether the membership includes professionals, members of the public or other stakeholders.

Blood services/suppliers and scientific organisations in the field of blood transfusion (that are often linked) were categorised as: yes, no, unclear. Examples are NHS Blood and Transplant, The British Blood Transfusion Society, The American Red Cross, The American Association of Blood Banks, the International Society of Blood Transfusion, the Deutsche Gesellschaft für Transfusionsmedizin und Immunhämatologie (German Blood Transfusion Society), the Société Française de Transfusion Sanguine (French Blood Transfusion Society),the Società Italiana di Medicina Transfusionale e Immunoematologia (Italian Blood Transfusion Society), the European Blood Alliance and the National Blood Authority Australia.

### Types of interventions

▶ Interventions targeting anaemia: pre-surgery iron therapy, perioperative cell salvage and autotransfusion and the use of restrictive red cell transfusion thresholds.

▶ Interventions targeting bleeding: tranexamic acid, point-of-care testing for coagulopathy.

### Controls

Participants not receiving the intervention, or alternative goal directed therapy.

### Outcomes

The primary transfusion outcome was exposure to red cell transfusion. The primary clinical outcome was 30-day or hospital all-cause mortality. Secondary outcomes included perioperative blood loss, re-operation for bleeding, numbers of red cells transfused, risk of receiving non-red cell components, acute brain injury (stroke, Transient Ischaemic Attack), myocardial infarction, low cardiac output, acute kidney injury stage 3 or requiring hemofiltration, sepsis and infection, intensive care unit and hospital length of stay, all as reported by study authors.

### Assessment of risk of bias in included studies

Included trials were appraised using the Cochrane risk of bias tool V.8.[16] Three authors (OF, ST, MR) assessed each outcome of interest as being at either low, high or unclear risk of bias for each domain. The adherence of trials to the CONSORT statement was also assessed.

### Data extraction

Data was extracted by three reviewers (OF, ST, MR) and managed using Microsoft Excel 2016 (Microsoft, Redmond, Washington, USA). This included number of authors, number of authors with declared conflicts of interest, year of publication, number of centres, number of participants, whether the study was designed to detect a treatment effect on clinical outcomes with the exclusion of transfusions, bleeding or use of healthcare resources and whether a primary outcome was specified. Cross validation of 10% of the selected studies was performed by the lead author (GM) to assess inter observer reproducibility. Excluded studies and the reason for exclusion were recorded.[17] Disagreements were resolved by discussion and consensus. In instances where this was not possible, the lead author (GM) determined whether or not the study was included.

### Data synthesis and measures of treatment effect

For dichotomous variables, the number of events in the treatment and control groups were collected, and the risk ratio (RR) with 95% CI was calculated. For continuous variables, the standardised mean difference with 95% CI was calculated. For the primary analysis, treatment effects for individual exposures of interest were estimated as RR (95% CI) using random effects models. All analyses were carried out using Review Manager (RevMan) V.5.4 (The Nordic Cochrane Centre, Copenhagen, Denmark), The Cochrane Collaboration, 2014.

### Dealing with heterogeneity

The $I^2$ statistic was used to estimate the percentage of total variation across studies attributed to heterogeneity, rather than chance.

### Subgroup analyses

Heterogeneity of treatment effects was explored using a pre-specified subgroup analysis for the following criteria: effects of Epoch—Prior to 2010 vs Post 2010 (to reflect widespread adoption of ICJME standards by editorial teams); ICJME statements in published text vs No ICJME statements; Country of origin for First Author (USA, Europe, Other).

### Sensitivity analysis

A pre-specified analysis was performed to assess Undeclared Author Conflicts of Interest. The authors of each manuscript were cross-checked between manuscripts for declared COIs. Where a COI had not been declared within 5 years of a declaration by that author in another trial, these were considered Undeclared Conflict of Interest. In the sensitivity analysis, the definition of Author Conflict of Interest was then recalibrated to include the revised classification and the analysis for the primary outcomes was repeated. A second sensitivity analysis was restricted to trials at low risk of bias.

### Reporting bias

Publication bias for the primary outcomes was assessed using funnel plots. Egger's test[18] was performed where there were 10 or more trials included in the analysis. The effects of reporting bias on the results of the primary analyses were assessed using trim and fill.[19]

### Patient and public involvement

Patients or the public were not involved in the design, or conduct, or reporting, or dissemination plans of our research.

## RESULTS

### Study selection

Searches identified 389 full-text publications reporting trials of five different PBM interventions enrolling 53 635 participants, for inclusion in the analysis (online supplemental eFigure 1). Eleven trials evaluated preoperative iron therapy (n=1031 participants), 42 trials evaluated autologous cell salvage and autotransfusion (n=5877), 22 trials compared restrictive versus liberal red cell transfusion thresholds (n=13 324), 298 trials evaluated tranexamic acid (n=32 496)

and 15 trials evaluated point-of-care tests for coagulopathic haemorrhage (n=907).

### Characteristics of included studies

The characteristics of included studies are presented in online supplemental eTable 1. Overall, 31 trials declared authorship COIs and 65 trials reported funding COIs. Of these, 16 studies had accessible ICMJE reporting statements.

### Risk of bias assessments

The summary of the risk of bias assessments is presented in online supplemental eFigure 2. Thirty-two studies (8%) were at low risk of bias in all domains, 265 (68%) were at low risk for selective reporting and 152 (39%) were at low risk of bias for allocation concealment.

### Data synthesis

Meta-analysis of all included trials showed that PBM interventions significantly reduced red cell transfusion RR 0.60, 95% CI 0.57 to 0.63, $I^2$=76%. Meta-analysis did not show significant treatment effects on mortality RR 0.93, 95% CI 0.81 to 1.07, $I^2$=0%. Assessment of reporting bias

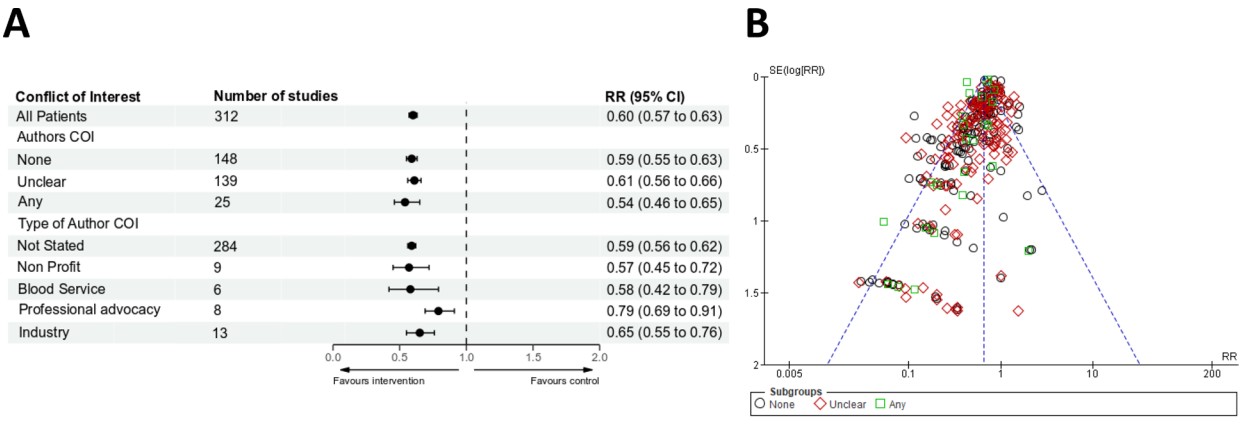

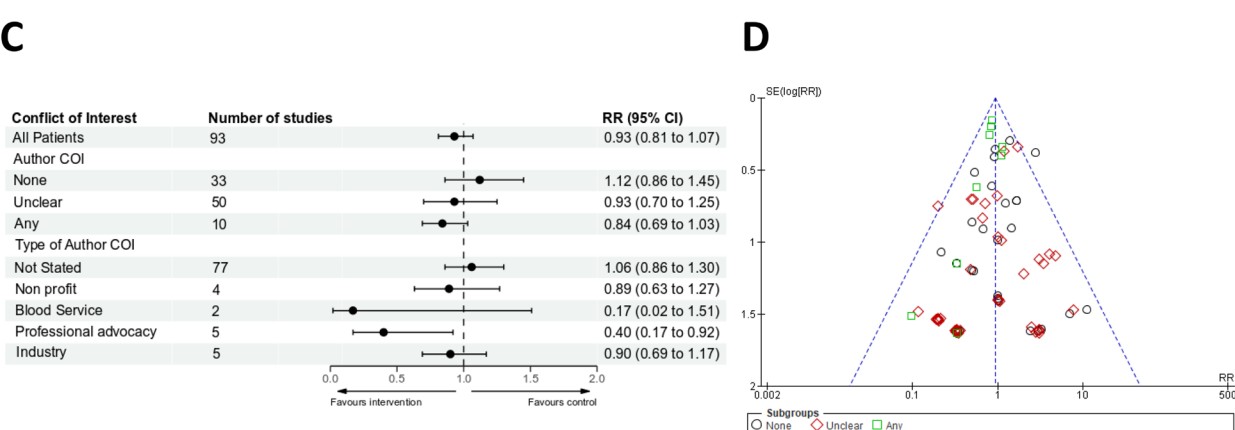

**Figure 1** (A) Forest plots for risk of receiving red cell transfusions based on Authors COI. Effects were expressed as risk ratios (RRs) with 95% CIs. (B) Funnel plots for risk of receiving red cell transfusions. There were insufficient numbers of trials to funnel plot each type of conflict of interest individually. (C) Forest plots for risk of mortality based on Authors COI. Effects were expressed as RRs with 95% CIs. (D) Funnel plots for risk of mortality. There were insufficient numbers of trials to funnel plot each type of conflict of interest individually. COI, conflict of interest.

using funnel plots demonstrated asymmetry for reported treatment effects on transfusion, but not for mortality (online supplemental eFigure 3).

## Author conflicts of interest on the co-primary outcomes

The risk of receiving red cell transfusion was assessed in 312 trials and was significantly reduced irrespective of whether an Author Conflicts of Interest, was declared, not declared or unclear, and with high heterogeneity (figure 1A). Funnel plots identified significant reporting bias (figure 1B). Trim and fill indicated that the effect of the bias favoured PBM interventions across all groups (online supplemental eFigure 3). The risk of transfusion was reduced irrespective of the type of COI (figure 1A).

30-day or hospital all-cause mortality was reported in 93 trials totalling 26766 patients. Eleven studies had no events reported in either group. In trials where there were no declared Author Conflicts of Interest, the treatment effect on 30-day or hospital all-cause mortality was RR 1.12, 95% CI 0.86 to 1.45, $I^2$=0%. In trials where Author Conflicts of Interest were declared, the treatment effect on mortality was RR 0.84, 95% CI 0.69 to 1.03, $I^2$=0%. In trials where Author Conflicts were unclear, the reported treatment effect on mortality was RR 1.06, 95% CI 0.86 to 1.3, $I^2$=0% (figure 1C). For mortality, funnel plot asymmetry was observed (p=0.04) in trials where authors had any declared conflicts of interest RR 0.85, 95% CI 0.71 to 1.02 (figure 1D). The results of trim and fill analysis, RR 0.92, 95% CI 0.72 to 1.17, indicated that the effect of the bias on the point estimate was towards the null (figure 2).

In trials where authors declared links to non-profit agencies, the estimated treatment effect on mortality was RR 0.89, 95% CI 0.63 to 1.27, $I^2$=0%. In trials where authors declared links to blood services, the treatment effect on mortality was RR 0.17, 95% CI 0.02 to 1.51, $I^2$=0%. In trials where authors declared links to industry, the treatment effect on mortality was RR 0.90, 95% CI 0.69 to 1.17, $I^2$=0%. In trials where authors were linked to professional advocacy organisations, the treatment effects

on mortality was RR 0.40, 95% CI 0.17 to 0.92, p=0.03, $I^2$=0% (figure 1C).

## Funding conflict of interest

The reduction in red cell transfusion rate attributable to PBM interventions was observed irrespective of whether any Funding conflicts were disclosed (figure 3A). Funnel plots and trim and fill indicated that there was reporting bias favouring PBM interventions (figure 3B). The observed reduction in transfusion was observed irrespective of the funding source (figure 3A).

In trials where no Funding Conflicts were declared, the treatment effect on mortality was RR 1.04, 95% CI 0.79 to 1.36, $I^2$=0%. In trials where a Funding Conflict was declared, the treatment effect on mortality was RR 0.84, 95% CI 0.69 to 1.02, $I^2$=0%. In trials where the Funding was unclear, the treatment effect on mortality was RR 1.04, 95% CI 0.79 to 1.39, $I^2$=0% (figure 3C). The assessment of funnel plots for asymmetry or trim and fill showed no significant difference for mortality based on funding COI (online supplemental eFigure 3, figure 3D).

In trials funded by non-profit agencies, the treatment effect on mortality was RR 0.95, 95% CI 0.76 to 1.19, $I^2$=0%. In trials funded by blood services, the treatment effect was RR 0.86, 95% CI 0.64 to 1.16, $I^2$=0%. In trials funded by industry, the treatment effect on mortality was RR 0.99, 95% CI 0.53 to 1.85, $I^2$=0%. In trials funded in whole or in part by professional advocacy organisations (four studies with 761 patients), the pooled treatment effect estimate on mortality was RR 0.40, 95% CI 0.17 to 0.96, $I^2$=0% (figure 3C).

## Secondary outcomes

All secondary outcome analyses were broadly consistent with the results of the primary analysis (online supplemental appendix eTable 2).

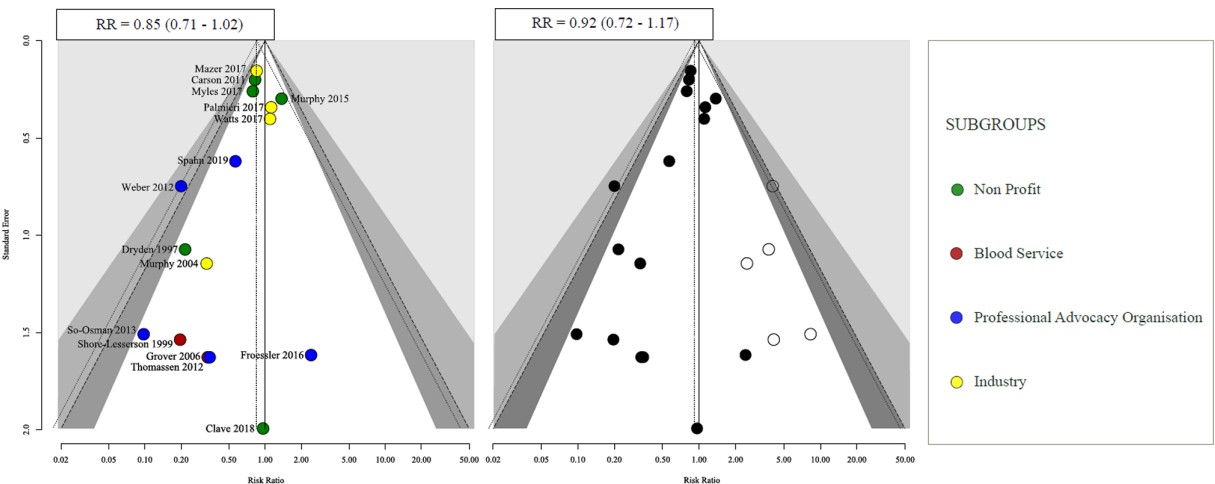

**Figure 2** Funnel plot (first figure) and trim and fill (second figure) obtained for mortality based on if any author conflicts of interest were present. RR, risk ratio.

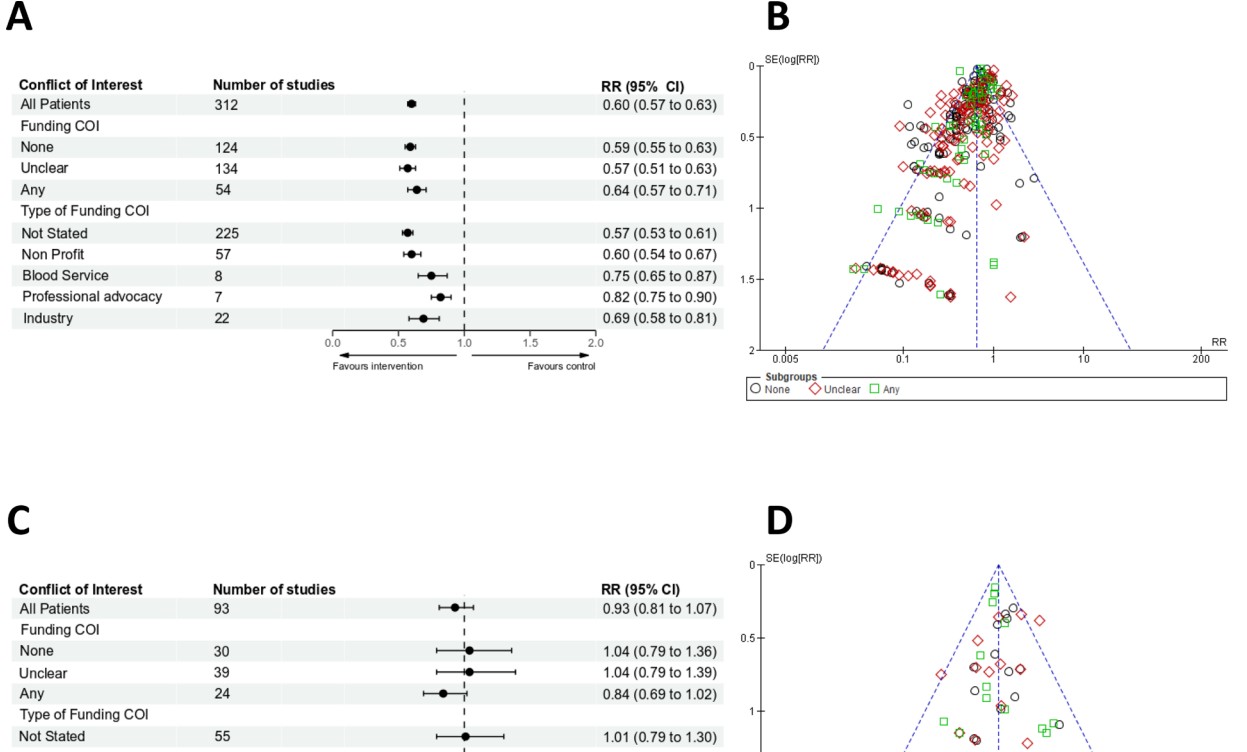

**Figure 3** (A) Forest plots for risk of receiving red cell transfusions based on Funding COI. Effects were expressed as risk ratios (RRs) with 95% CIs. (B) Funnel plots for risk of receiving red cell transfusions. There were insufficient numbers of trials to funnel plot each type of conflict of interest individually. (C) Forest plots for risk of mortality based on Funding COI. Effects were expressed as RRs with 95% CIs. (D) Funnel plots for risk of mortality. There were insufficient numbers of trials to funnel plot each type of conflict of interest individually. COI, conflict of interest.

## Subgroup analyses

In a pre-specified subgroup analysis, we hypothesised that reporting bias for clinical outcomes would be more likely for trials where these were secondary outcomes, versus trials where these were primary outcomes, as observed in larger higher quality trials. For trials where the primary outcome was a clinical event, the pooled treatment effect estimate for mortality was RR 1.14, 95% CI 0.88 to 1.49, $I^2$=25%. For trials where the primary outcome was not a clinical event, the pooled treatment effect estimate for mortality was RR 0.81, 95% CI 0.66 to 1, $I^2$=0%, P for overall effect 0.34, P value for interaction was 0.04 (online supplemental eTable 3).

There was no significant interaction between the country origin of the corresponding author (online supplemental eTable 4). Sixteen studies had ICMJE reporting statements. There was no significant interaction between journal publications that adhered to the ICMJE standards for reporting conflicts of interest and those that did not for the primary outcomes (online supplemental eTable 5). There was no significant interaction between studies published before or after 2010 for mortality or risk of red cell transfusions (online supplemental eTable 6).

## Sensitivity analysis

Repeating the primary analysis after reclassifying 17 trials where authors were considered to have undeclared conflicts of interest (online supplemental eTable 7) did not change the overall results (online supplemental eTable 8). When studies at high or unclear risk of selection bias were excluded, mortality was significantly reduced (RR 0.4 95% CI 0.17 to 0.92, $I^2$=0%, p=0.03) where authors had conflicts of interest related to professional advocacy organisations, whereas the risk of red cell transfusions was significantly reduced irrespective of any declared COI (online supplemental eTable 9).

## DISCUSSION
### Main findings

In a systematic review of RCTs, we have previously demonstrated that PBM interventions reduce red cell transfusion but have little or no treatment effect on mortality or other important clinical outcomes in people undergoing major surgery. This secondary analysis has provided further insights into these observations. These results

clearly show that: (1) the evidence indicates that PBM interventions reduce transfusion. (2) Funnel plots and Egger's tests are highly suggestive of reporting bias. (3) Fill and trim demonstrated that the reporting bias was in favour of the treatment effects of PBM on reducing transfusion. We therefore interpret these results as showing clear links between reporting bias and the magnitude of the treatment effect on transfusion, one of our primary endpoints. First, we observed reporting bias in favour of the treatment effects of PBM interventions on transfusion (Funnel plots and trim and fill in 312 studies and 56 686 patients). Second, we observed that treatment effects on mortality favoured PBM interventions where authors had declared conflicts of interest, with evidence of reporting bias (Funnel plots and trim and fill in 16 studies and 16 077 patients). This was not observed in trials with no reported conflicts. Third, we observed that trials where authors had declared links to professional PBM advocacy organisations reported statistically significant reductions in mortality, unlike other groups (Forest plot in five studies and 977 patients). Fourth, we observed that overall treatment effects on mortality tended to favour PBM interventions in trials with a potential Funding conflict. Specifically, trials funded in whole or in part by professional PBM advocacy organisations reported statistically significant reductions in mortality, unlike other groups (Forest plot in four studies and 761 patients) Fifth, the results of the primary analysis were consistent across a range of secondary and sensitivity analyses (Subgroup analysis with 93 studies and 26 766 patients for mortality, 312 studies and 55 546 for risk of red cell transfusion and sensitivity analysis for low allocation bias with 51 studies and 20 973 patients for mortality, 133 studies and 30 169 patients for risk of red cell transfusion).

Our secondary outcomes analyses demonstrated (online supplemental eTable 2) heterogeneity in disease definitions, reported outcomes and estimated treatment effects. The definition of adverse events in particular was very heterogeneous between studies, limiting assessment of this data. Overall, 8/102 secondary outcome analyses for important clinical outcomes stratified by type of conflict yielded a p value for treatment effect <0.05. Analyses of bleeding and transfusion outcomes generally favoured PBM, as per the findings of our primary analysis of red cell transfusion.

## Clinical importance

Red cell transfusion is one of the most commonly used interventions in hospitalised patients, with over 2.5 million red cell units transfused in the UK per year.[20] Donated blood is a precious resource. Steps to minimise transfusion are welcome, and indeed necessary in situations where there are concerns about the blood supply. PBM has been recently defined as a patient-centred, systematic, evidence-based approach to improve patient outcomes by managing and preserving a patient's own blood, while promoting patient safety and empowerment.[21] Recent guidelines advocate the implementation of multiple interventions to prevent the use of blood, on the basis that this results in improved outcomes for patients or cost effectiveness.[2] The current analysis which included 389 studies in 53 635 patients adds further uncertainty as to whether PBM interventions have important clinical benefits. First, the evidence suggests that that the effects of PBM on transfusion are less than estimated from trial data, due to reporting bias. This occurred even in trials where no conflicts of interest were reported. The multiple potential sources of bias identified in included RCTs, including increased risk of selection bias (68%), lack of blinding (67%) and reporting bias (61%), as well as unmeasured conflicts,[22–24] may have contributed to these results.

Second, RCTs linked to PBM advocacy organisations reported significant clinical benefits, unlike other identified sources of COI. The reasons for this are unclear from the data. Professional PBM advocacy organisations are typically composed of clinicians who advocate for the implementation of PBM interventions in the belief that the benefits of these outweigh the risk. As a result, they are strong drivers for change.[25–27] They also have poorly defined links to industry.[14 16 28 29] These potential sources of bias, unconscious or otherwise, can influence trial design, management and reporting.[29] Along with the methodological limitations identified in the majority of the trials, we conclude that the quality of the evidence used to inform PBM decisions poor. The results identify an unmet need for better quality trials, free of conflicts or where conflicts are appropriately managed, to establish appropriate indications for PBM. This is difficult, given that international PBM guidelines have already been published,[2] and PBM is being rapidly implemented in many health systems, including in the NHS, often led by professional PBM advocacy groups and consultancies. Nonetheless, the current study provides further evidence that better trials are needed.

## Strengths and limitations

The study has important strengths. First, it is the most comprehensive review of PBM RCTs in people undergoing surgery to date. Second, it used Cochrane methodology, objective measures for the co-primary outcomes that would be consistent across trials and settings, and was reported against a pre-specified and registered protocol. Third, despite the multiple settings and interventions there was very little heterogeneity in the estimates of the treatment effects on clinical outcomes. This consistency is further evidence that PBM has little or no impact on clinical outcomes. The study has important limitations. First, the low methodological quality of many of the studies lowers certainty as to the precision of the estimates of treatment effect on primary and secondary outcomes, although similar treatment effects were observed when the analysis was restricted to groups at low risk of important bias, or in larger trials designed to detect differences in important clinical outcomes. Second, we relied on self-reported conflicts of interest in published trial reports for the primary analyses. Journal adherence to declarations of conflicts improved after the introduction of ICMJE reporting

standards, which provides an international consensus framework for assessing and reporting conflicts, however, these standards were present only in a minority of trials. It is therefore possible that undeclared conflicts may have altered our results. We addressed this by comparing the effect of epoch (publication before or after 2010 on outcomes), as ICJME standards were almost ubiquitous after this time. No significant interaction was observed. We also attempted to adjust for undeclared conflicts, measured against pre-specified criteria, however, this only identified a small number of trials with potentially undeclared conflicts (17/389, 4%). Given the changes in reporting standards over the time period covered by the review it is not certain how specific or sensitive this definition may have been. Third, the numbers of trials with conflicts linked to PBM advocacy organisations was low, and we cannot exclude that treatment estimates may change with the addition of a small number of additional trials. From the four studies with funding linked to PBM advocacy organisation reporting mortality, two investigated the use of iron and two point of care testing. We acknowledge that the analysis is unable to measure the direct influence of PBM advocacy groups on trial conduct and reporting. These trials also evaluated different PBM interventions, although we have previously reported this is unlikely to have contributed to heterogeneity with respect to clinical outcomes; all five PBM interventions evaluated in a previous review had little or no effect on important clinical outcomes.[3] Fourth, the majority of the studies included in the secondary analysis were not designed to assess the impact of PBM measures on mortality. Fifth, the last searches in the primary analysis were completed in June 2019, with recent high quality studies published after this date not being included in the analysis. Finally, the review omitted RCTs in obstetrics, trauma (including neurosurgery) and gynaecology from the analyses. This raises the possibility of selection bias in our sample. In mitigation, we have performed the largest and most comprehensive review of PBM interventions thus far reported, updating relevant Cochrane reviews including all the data on these interventions used in contemporary treatment guidelines and strengthened by recent evidence.[3 10–14 30 31] We therefore consider the sample to be representative of the evidence used to guide PBM decisions in most surgical settings.

In conclusion, a secondary analysis of a systematic review of RCTs of PBM interventions in people requiring surgery has identified further limitations in the evidence to support PBM, specifically reporting bias that acts to favour PBM, and evidence that trials undertaken by some groups report clinical benefits that are not observed in groups without similar conflicts. These results caution against the widespread introduction of PBM without better evidence, and highlight the need for further research in this area.

**Contributors** All authors had full access to all of the data in the study and take responsibility for the integrity of the data and the accuracy of the data analysis. Study concept and design: GM/MR. Acquisition of data: MR/OF/ST.Analysis and interpretation of data: MR/OF/ST/RGA/FL/TR/GM.Drafting of the manuscript: MR/RA/OF/ST/FL/TR/GM.Study supervision: GM. GM is the author responsible for the overall content as the guarantor.

**Funding** GM and YL are supported by British Heart Foundation grant CH/12/1/29419. MR is supported by the National Institute for Health and Care Research award CL-2020-11-003. The funder had no role in study design, data collection, analysis, or interpretation, or writing of the report. The corresponding author had full access to all the data in the study and had final responsibility for the decision to submit for publication.

**Competing interests** GM reports grants from the British Heart Foundation during the conduct of the study, and grants from Zimmer Biomet. GM reports support for educational activities from Terumo, outside the submitted work. TR reports grants from UK, NIHR HTA, grants from Australian, NHMRC, grants, personal fees and non-financial support from Pharmocosmos, grants, personal fees and non-financial support from Vifor Pharma, grants from UK, NIHR EME, grants from Australian MRFF, grants from Western Australia FHRF, grants and personal fees from Pfizer Australia, personal fees from BioAge Labs, outside the submitted work; and TR is a regular speaker at national and international conferences on anaemia, blood transfusion, wound healing and vascular diseases for which he has received expenses for travel, accommodation and sundries. TR has worked with several agencies promoting meetings or healthcare. TR is a director of The Iron Clinic Ltd and director of Veincare London Ltd & Veincare WA also TR is the Vascular lead for 18-week wait Ltd. None of these conflicts of interest have any direct relationship or influence on the manuscript presented. No conflicts of interest relevant to this manuscript were disclosed by the reviewers or editor. The authors are unable to assess the sources of bias associated with the reviewers or editor in the open peer review process.

**Patient and public involvement** Patients and/or the public were not involved in the design, or conduct, or reporting, or dissemination plans of this research.

**Patient consent for publication** Not applicable.

**Provenance and peer review** Not commissioned; externally peer reviewed.

**Data availability statement** Data are available upon reasonable request. Additional raw data, including the RevMan files can be shared by requests submitted to the corresponding author's email.

**ORCID iDs**
Marius Roman http://orcid.org/0000-0003-2269-0500
Florence Lai http://orcid.org/0000-0003-4354-0624

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
