## [Reviewer comments · BMJ Open]

ARTICLE DETAILS

TITLE (PROVISIONAL)	Reporting Conflicts of Interest in randomised trials of Patient Blood Management interventions in patients requiring major surgery: A Systematic review and Meta-analysis
AUTHORS	Roman, Marius; Fashina, Oluwatomini; Tomassini, Sara; Abbasciano, Riccardo; Lai, Florence; Richards, Toby; Murphy, Gavin

VERSION 1 – REVIEW

REVIEWER	Mahmoud Loubani Hull University Teaching Hospitals NHS Trust, Cardiothoracic Surgery
REVIEW RETURNED	10-Sep-2021

GENERAL COMMENTS	This paper addresses a sensitive topic with scientific approach. The paper is well written with clear conclusions
---

REVIEWER	Suma Choorapoikayil Forschungszentrum Europäische Aufklärung
REVIEW RETURNED	16-Nov-2021

GENERAL COMMENTS	Roman and colleagues performed a secondary analysis of a meta-analysis published in 2020 (Roman MA, et al. Patient blood management interventions do not lead to important clinical benefits or cost-effectiveness for major surgery: a network meta-analysis. BJA. 2020). The submitted work to BMJ Open aimed to systematically review the effects of declared and undeclared conflicts of interest on RCTs of Patient Blood Management (PBM) interventions. Taken together, the authors performed an extensive analysis and revealed that: 1) PBM interventions significantly reduced RBC transfusion (RR 0.60, 95% CI 0.57-0.63) (page 11, line 40)2) PBM interventions did not increase risk for mortality (RR 0.93, 95% CI 0.81-1.07) (page 11, line 42) Furthermore, considering Authors Conflict of Interest the authors revealed that: 3) Irrespective of whether an Authors Conflict of Interest was declared-not declared-or-unclear, PBM interventions significantly reduced RBC transfusion (Figure 1A) (page 11, line 59)4) Irrespective of whether an Authors Conflict of Interest was declared-not declared-or-unclear, PBM interventions did not increase risk for mortality (Figure 1C) (page 12, line 9) In addition, considering Funding Conflict of Interest the authors revealed that: 3) Irrespective of whether any Funding Conflicts of Interest were disclosed, PBM interventions significantly reduced RBC transfusion
---

	(Figure 2A) (page 12, line 39) 4) Irrespective of whether any Funding Conflicts of Interest were disclosed, PBM interventions did not increase risk for mortality (Figure 2C) (page 12, line 49-51) Finally, the authors demonstrated that: 5) Irrespective of whether an Authors Conflict of Interest was declared-not declared-or-unclear OR any Funding Conflicts of Interest were disclosed, PBM interventions did not increase risk for myocardial infarction rate, risk for acute kidney injury stage 3/ acute brain injury/ sepsis and infection (eTable2) 6) Irrespective of whether an Authors Conflict of Interest was declared-not declared-or-unclear OR any Funding Conflicts of Interest were disclosed, PBM interventions significantly reduced the number of transfused RBCs (eTable2) 7) Irrespective of whether an Authors Conflict of Interest was declared-not declared-or-unclear OR any Funding Conflicts of Interest were disclosed, PBM interventions significantly reduced perioperative blood loss (eTable2) 8) Irrespective of whether an Authors Conflict of Interest was declared-not declared-or-unclear OR any Funding Conflicts of Interest were disclosed, PBM interventions significantly reduced risk of receiving FFP (eTable2) The authors performed additional sensitivity analysis and found no robust effect size estimates for the endpoints: adverse reaction, low cardiac output, reoperation for bleeding, risk of receiving platelets, intensive care length of stay, and hospital length of stay indicating that further studies are needed to assess the impact of PBM for these clinical important endpoints (eTable2). In summary, the analysis revealed that -irrespective of whether an Authors Conflict of Interest was declared-not declared-or-unclear OR any Funding Conflicts of Interest were disclosed- PBM significantly reduce RBC transfusions and perioperative blood loss and did not increase mortality rate, myocardial infarction rate, risk for acute kidney injury stage 3, acute brain injury, sepsis or infection. However, from my opinion of view, the content of the discussion (and also of the abstract) is not in line with the results: The authors claim in the discussion section (page 14, line 13-33) "First, we observed reporting bias in favour of the treatment effects of PBM interventions on transfusion." -> which, however, as demonstrated by the authors had no significant impact on the investigated outcome data as demonstrated in the result section. "Second we observed that treatment effects on mortality favoured PBM interventions where authors had declared conflicts of interest, with evidence of reporting bias. This was not observed in trials where there no reported conflicts." -> which, however, as demonstrated by the authors had no significant impact on the investigated outcome data as demonstrated in the result section. "Third we observed that trials where authors had declared links to professional PBM advocacy organisations reported statistically significant reductions in mortality, unlike other groups." -> This conclusion is based on the analysis of 4 studies including 761 patients and RR 0.4 (95%CI 0.17-0.96); p<0.04). However, the authors do not explain whether the 4 studies analysed the same PBM intervention and do not reveal the direct influence of the PBM advocacy organisation on the study design and analysis. "Fourth, we observed that overall treatment effects on mortality tended to favour PBM interventions in trials with a potential Funding conflict. Specifically, trials funded in whole or in part by professional PBM advocacy organisations reported statistically significant reductions in mortality, unlike other groups. " -> could the authors
--	--

	indicate which of their analysis showed a significant impact of PBM interventions on mortality? “Fifth, the results of the primary analysis were consistent across a range of secondary and sensitivity analyses.” -> Yes, but in the context of the above mentioned statements this sentence is very misleading. While this is an extensive work, the authors aimed to find an association between Conflict of Interest/Funding disclosure and outcome data and have rather over-simplified their conclusions that are not based on their results. In addition, I have several major concerns:  • PBM consist of approximately 100 different measures and the authors evaluated 5. Thus, a relevant selection bias is present. • The authors used RR to measure effect size. What about studies with zero events? • In total 389 studies including 53,635 patients assessing the impact of 5 PBM interventions. Thus, a mean of 137.9 patients per study were included and analysed. • Which tool was used to categorize conflict of interest? • Why were studies performed in obstetrics and neuro surgery excluded from analysis? Particularly, within the last decades several studies have been performed focusing on several PBM interventions in these patient groups. Again, a relevant selection bias is present? • The analysis included studies published till 1st June 2019. This secondary analysis is not timely, as several (high ranking) studies have been published recently. • Within clinical trials various people contribute to data acquisition, analysis and interpretation of which the majority receive departmental funding and should therefore not possess any bias. Thus, the impact of Funding or Authors Conflicts on outcome data should be judged with caution and in more detail. • The risk of bias assessment shown in eFigure2 is misleading as no distinction was made for the investigated PBM interventions. Here, risk of bias for allocation is judged with high risk more often in studies using cell saver compared to studies using TXA for example. • Out of 389 included studies, only 93 trials were included to assess the impact of PBM interventions on mortality. No information is provided about the total number of patients included in the analysis. How many of the 93 trials showed zero events in both study arms and were excluded from the RR analysis? • I assume a typo is present in the p-value of the following sentence? “For mortality, funnel plot asymmetry was observed in trials where authors had Any declared conflicts of interest RR 0.85, 95% CI 0.71-1.02, p=0.04” • Please rephrase the sentence “Patient Blood Management moves this one step further, advocating the implementation of multiple interventions to prevent the use of blood, on the basis that this results in improved outcomes for patients or cost effectiveness (page 14, line 42-48)”. PBM is a multidisciplinary approach to provide a comprehensive anaemia management, to reduce surgical blood loss and to provide guideline adherent transfusion practise. • Furthermore, there are several sentences in the discussion referring to articles which could cause misleading interpretation. • More importantly, how do the authors proof that the conflict of interest/funding conflicts stated in the “Conflict of interest statement” of each author had no influence on the outcome of this study? Overall, I do not recommend the publication of the submitted work in BMJ Open and hope that my comments will be helpful for the authors.
--	---

VERSION 1 – AUTHOR RESPONSE

Referee: 1

Comments to the Author

This paper addresses a sensitive topic with scientific approach. The paper is well written with clear conclusions.

Answer: Thank you for this statement.

Referee: 2

Comments to the Author

Taken together, the authors performed an extensive analysis

Answer: Thank you for the acknowledgement of the comprehensive approach to answer an important research question.

The authors performed additional sensitivity analysis and found no robust effect size estimates for the endpoints: adverse reaction, low cardiac output, reoperation for bleeding, risk of receiving platelets, intensive care length of stay, and hospital length of stay indicating that further studies are needed to assess the impact of PBM for these clinical important endpoints (eTable2)

Answer: Thank you for this comment. As suggested by the reviewer, in the revised discussion we acknowledge that the low quality of the included trials leads to important uncertainty as to the treatment effects of PBM on all outcomes in the manuscript, and the need for further research (Page 13 Paragraph 1).

In summary, the analysis revealed that -irrespective of whether an Authors Conflict of Interest was declared-not declared-or-unclear OR any Funding Conflicts of Interest were disclosed- PBM significantly reduce RBC transfusions and perioperative blood loss and did not increase mortality rate, myocardial infarction rate, risk for acute kidney injury stage 3, acute brain injury, sepsis or infection.

Answer: We agree with the reviewer. This is stated clearly in the manuscript (Page 13 Paragraph 2). One important caveat is that treatment effect estimates for continuous secondary outcomes; bleeding, numbers of units transfused, length of ICU and hospital stay, were characterised by high heterogeneity in almost every analyses, making the point estimates unreliable. It is therefore incorrect to claim that the data show a treatment effect on these outcomes.

The content of the discussion (and also of the abstract) is not in line with the results: The authors claim in the discussion section (page 14, line 13-33) “First, we observed reporting bias in favour of the treatment effects of PBM interventions on transfusion.” -> which, however, as demonstrated by the authors had no significant impact on the investigated outcome data as demonstrated in the result section.

Answer: We politely disagree with the reviewer on this point. The statement paraphrased from the Main findings section of the discussion is based on the evidence of reporting bias on the risk of blood transfusion in the funnel plots analysis as described in the last paragraph on page 9 Paragraph 5: ‘Funnel plots identified significant reporting bias (Figure 1B). Trim and fill indicated that the effect of the bias favoured PBM interventions across all groups (eFigure 3).’

These results clearly show that:

1. The evidence indicates that PBM interventions reduce transfusion.
2. Funnel plots and Egger’s tests are highly suggestive of reporting bias.
3. Fill and trim demonstrated that the reporting bias was in favour of the treatment effects of PBM on reducing transfusion.

We therefore interpret these results as showing clear links between reporting bias and the magnitude of the treatment effect on transfusion, one of our primary endpoints.

“Second we observed that treatment effects on mortality favoured PBM interventions where authors had declared conflicts of interest, with evidence of reporting bias. This was not

observed in trials where there no reported conflicts.” -> which, however, as demonstrated by the authors had no significant impact on the investigated outcome data as demonstrated in the result section.

Answer: We respond to this point in the same manner as we responded to the point above. This statement in the discussion is based on the demonstration of reporting bias on mortality as described in the results section (Page 10 Paragraph 1): “For mortality, funnel plot asymmetry was observed in trials where authors had Any declared conflicts of interest RR 0.85, 95% CI 0.71-1.02, p=0.04 (**Figure 1D**). The results of trim and fill analysis RR 0.92, 95% CI 0.72-1.17 indicated that the effect of the bias was to favour PBM interventions (**eFigure 3**).”

We feel that these results could only be interpreted as showing a relationship between reporting bias and outcomes.

“Third we observed that trials where authors had declared links to professional PBM advocacy organisations reported statistically significant reductions in mortality, unlike other groups.” -> This conclusion is based on the analysis of 4 studies including 761 patients and RR 0.4 (95%CI 0.17-0.96); p<0.04). However, the authors do not explain whether the 4 studies analysed the same PBM intervention and do not reveal the direct influence of the PBM advocacy organisation on the study design and analysis.

Answer: Thank you for highlighting this important result of our manuscript.

- We acknowledge the small numbers of trials in this analysis as a limitation of the discussion (Page 14 Paragraph 1).
- In the revision, we also acknowledge that these trials evaluated different PBM interventions (Page 14 Paragraph 1). However we draw attention to the observation in a previous review of this data (doi: 10.1016/j.bja.2020.04.087) that the treatment effect of PBM on clinical outcomes is consistent across type of intervention (consistently no effect), and therefore this is unlikely to have had an influence on our results or conclusions.
- Finally, we acknowledge that the analysis is unable to measure the direct influence of PBM advocacy groups on trial conduct and reporting. This is difficult to measure. For this reason, we used the ICMJE reporting standards against which individual trial reports were assessed. Whilst this cannot point to specifics, it provides an international consensus framework for assessing and reporting conflicts. This approach demonstrated the effects of these conflicts on trial reporting in a pre-specified, transparent, verifiable, and unambiguous way.

“Fourth, we observed that overall treatment effects on mortality tended to favour PBM interventions in trials with a potential Funding conflict. Specifically, trials funded in whole or in part by professional PBM advocacy organisations reported statistically significant reductions in mortality, unlike other groups. “ -> could the authors indicate which of their analysis showed a significant impact of PBM interventions on mortality?

Answer: Thank you for this comment. The results to support this statement are as follows:
Page 10 Paragraph 1. In trials where authors were linked to professional advocacy organisations the treatment effects on mortality was RR 0.40, 95% CI 0.17-0.92, P=0.03, I²=0% (Figure 1C).
Page 11. Paragraph 1. In trials funded in whole or in part by professional advocacy organisations the pooled treatment effect estimate on mortality was RR 0.40, 95% CI 0.17-0.96, I²=0%. (Figure 2C)

“Fifth, the results of the primary analysis were consistent across a range of secondary and sensitivity analyses.” -> Yes, but in the context of the above mentioned statements this sentence is very misleading.

While this is an extensive work, the authors aimed to find an association between Conflict of Interest/Funding disclosure and outcome data and have rather over-simplified their conclusions that are not based on their results.

Answer: Thank you for this comment. We selected our co-primary outcomes as these are objective, verifiable outcomes resistant to detection (but not reporting) bias. Our secondary outcomes did not have these qualities (eTable 2 in the Supplement) with heterogeneity in disease definitions, reported outcomes, and estimated treatment effects. The definition of Adverse Events in particular was very heterogeneous between studies, limiting assessment of this data. Overall, 8/102 secondary outcome analyses for important clinical outcomes stratified by type of conflict yielded a p value for treatment effect <0.05. Analyses of bleeding and transfusion outcomes generally favoured PBM, as per the

findings of our primary analysis of red cell transfusion. For these reasons, we believe that our statement in the results section (Page 11, Paragraph 2) that 'All secondary outcome analyses were broadly consistent with the results of the primary analysis' is accurate.

PBM consist of approximately 100 different measures and the authors evaluated 5. Thus, a relevant selection bias is present.

Answer: Thank you for this comment. In our revision, we have acknowledged this as a limitation in the discussion section (Page 14, paragraph 1).

In mitigation however, the 5 PBM interventions that we identified are in common use and have been evaluated in 389 RCTs, Cochrane systematic reviews (doi: 10.1002/14651858.CD002042.pub4, doi: 10.1002/14651858.CD001888.pub3doi: 10.1002/14651858.CD007871.pub3) and evidence based treatment guidelines (<https://www.nice.org.uk/guidance/ng24>, doi: 10.1053/j.jvca.2017.06.026., DOI: 10.1007/s00134-021-06531-x). The review was comprehensive, and we believe includes the great majority of evidence in use to guide clinical practice. Importantly, it includes all the relevant data from the recent International Consensus PBM guidelines, to which the Lead author GJM contributed; doi: 10.1001/jama.2019.0554. We therefore suggest that this is a representative sample of the evidence used to guide most PBM decisions.

The authors used RR to measure effect size. What about studies with zero events?

Answer: The statistical analysis was performed in Revman, which is the requirement for Cochrane reviews. Studies with zero events in both groups are not considered in a relative risk calculation.

In total 389 studies including 53,635 patients assessing the impact of 5 PBM interventions. Thus, a mean of 137.9 patients per study were included and analysed.

Answer: This is correct.

Which tool was used to categorize conflict of interest?

Answer: Conflicts of interest were assessed for individual trials based on the International Committee of Medical Journal Editors (ICMJE) standards for reporting conflicts of interest. This has now been clarified in the Methods section.

Why were studies performed in obstetrics and neuro surgery excluded from analysis? Particularly, within the last decades several studies have been performed focusing on several PBM interventions in these patient groups. Again, a relevant selection bias is present?

Answer: This is a welcome point of discussion. These studies were excluded from our original analysis and therefore excluded from this secondary analysis. This is listed as a potential limitation in the discussion (Page 14, Paragraph 1).

The analysis included studies published till 1st June 2019. This secondary analysis is not timely, as several (high ranking) studies have been published recently.

Answer: Thank you for this comment. The paper has been in review since June 2021. We acknowledge that the searches finished in June 2019 and would consider an updated search. However, this analysis is substantial and outcomes related to conflicts of interest are unlikely to change significantly with time as demonstrated in the Epoch sensitivity analysis.

Within clinical trials various people contribute to data acquisition, analysis and interpretation of which the majority receive departmental funding and should therefore not possess any bias. Thus, the impact of Funding or Authors Conflicts on outcome data should be judged with caution and in more detail.

Answer: Thank you for this discussion. We agree with the reviewer. It is for this reason that we use the ICMJE reporting standards against which individual trial reports were assessed. Whilst this cannot point to specifics, it provides an international consensus framework for assessing and reporting

conflicts. This also means that the analysis is restricted to a description of the effects of these conflicts on reported outcomes.

The risk of bias assessment shown in eFigure2 is misleading as no distinction was made for the investigated PBM interventions. Here, risk of bias for allocation is judged with high risk more often in studies using cell saver compared to studies using TXA for example.

Answer: The reviewer is correct in the remark that this is an overall view of the risk of bias. This information is presented in the original network meta-analysis. (<https://doi.org/10.1016/j.bja.2020.04.087>) The percentage of studies at high risk for allocation bias in the cell salvage studies is 18% (8 studies), while for tranexamic acid is 12.7% (39 studies). This difference may be explained by the difference in the logistics for blinding equipment compared to medication interventions.

Out of 389 included studies, only 93 trials were included to assess the impact of PBM interventions on mortality. No information is provided about the total number of patients included in the analysis. How many of the 93 trials showed zero events in both study arms and were excluded from the RR analysis?

Answer: Thank you for this comment. This data has now been included in the results section.

I assume a typo is present in the p-value of the following sentence? “For mortality, funnel plot asymmetry was observed in trials where authors had Any declared conflicts of interest RR 0.85, 95% CI 0.71-1.02, p=0.04”

Answer: Thank you for pointing this remark. The p value is for the Egger’s test analysis. The sentence has now been rephrased to “For mortality, funnel plot asymmetry was observed (p=0.04) in trials where authors had Any declared conflicts of interest RR 0.85, 95% CI 0.71-1.02 (Figure 1D).”

Please rephrase the sentence “Patient Blood Management moves this one step further, advocating the implementation of multiple interventions to prevent the use of blood, on the basis that this results in improved outcomes for patients or cost effectiveness (page 14, line 42-48)”. PBM is a multidisciplinary approach to provide a comprehensive anaemia management, to reduce surgical blood loss and to provide guideline adherent transfusion practise.

Answer: Thank you for this comment. We politely disagree. PBM was intended *a priori* to demonstrate the effects of blood conservation interventions on clinically important outcomes. This is clearly stated in the most recent international PBM consensus guidelines (doi: 10.1001/jama.2019.0554) which quote directly from the World Health Organisation definition of PBM as “a patient-focused, evidence-based and systematic approach to optimize the management of patients and transfusion of blood products for quality and effective patient care. *It is designed to improve patient outcomes through the safe and rational use of blood and blood products and by minimizing unnecessary exposure to blood products...*” https://www.who.int/bloodsafety/events/gfbs_01_pbm/en/

This review raises legitimate questions as to whether the published evidence supports the claim that PBM has clinical benefits. Worryingly, the current report highlights inconsistencies in the reporting of PBM trials across different types of Conflict of Interest. This finding is robust to sensitivity analyses, the evidence presented is clear, and these results are of interest to a wide audience of clinicians and researchers.

Furthermore, there are several sentences in the discussion referring to articles which could cause misleading interpretation.

Answer: Thank you for this comment. We are unclear which specific articles are referred to in this statement. However, we have made significant revisions to the discussion and we hope that these are satisfactory.

More importantly, how do the authors proof that the conflict of interest/funding conflicts stated in the “Conflict of interest statement” of each author had no influence on the outcome of this study?

Answer: As discussed to a previous point raised by the reviewer, we do not have evidence as to how individual conflicts may have influenced the design, conduct, interpretation, or reporting of trials. However, using the internationally agreed ICMJE framework for the reporting of conflicts of interest in randomised trials we have shown that stakeholders grouped by the nature of their *self-reported* conflicts of interest show evidence of reporting bias favouring PBM interventions, as well as statistically significantly different results in trials, in important groups favouring PBM interventions.

VERSION 2 – REVIEW

REVIEWER	Suma Choorapoikayil Forschungszentrum Europäische Aufklärung
REVIEW RETURNED	08-Mar-2022

GENERAL COMMENTS	Thank you for answering my questions, and for the adaptations made. I consider that the revisions improved considerably the text. However, I have several issues which have not been adequately addressed in the rebuttal letter and revised manuscript: 1) The authors performed an extensive analysis with several important results. To enhance the transparency and prevent any reporting bias I suggest to include to each discussed conclusion the corresponding result: which analysis was performed, the number of analysed studies and patients. 2) Thank you very much for clarifying the line of thoughts for the conclusion stated in the first paragraph on page 14 (page 14, lines 15-35). Please include the following statement in line 15/page 14: “These results clearly show that: 1. The evidence indicates that PBM interventions reduce transfusion. 2. Funnel plots and Egger’s tests are highly suggestive of reporting bias. 3. Fill and trim demonstrated that the reporting bias was in favour of the treatment effects of PBM on reducing transfusion. We therefore interpret these results as showing clear links between reporting bias and the magnitude of the treatment effect on transfusion, one of our primary endpoints.” 3) The authors found that trials in which authors had declared links to professional PBM advocacy organisations reported statistically significant reductions in mortality, unlike other groups. This conclusion is based on the analysis of 4 studies including 761 patients and RR 0.4 (95%CI 0.17-0.96); $p < 0.04$). Please include this information to the corresponding section in the discussion and provide information whether the 4 studies analysed the same PBM intervention. As stated in your rebuttal letter, include also the following information: we acknowledge that the analysis is unable to measure the direct influence of PBM advocacy groups on trial conduct and reporting. 4) Include the following statement from your rebuttal letter to the discussion section: “Our secondary outcomes did not have these qualities (eTable 2 in the Supplement) with heterogeneity in disease definitions, reported outcomes, and estimated treatment effects. The definition of adverse events in particular was very heterogeneous between studies, limiting assessment of this data. Overall, 8/102 secondary outcome analyses for important clinical outcomes stratified by type of conflict yielded a p value for treatment effect < 0.05. Analyses of bleeding and transfusion outcomes generally favoured PBM, as per the findings of our primary analysis of red cell transfusion.”
--

	5) PBM is a multimodal concept consisting of several measures that in combination enhance patient safety. Studies included in the secondary analysis were not designed to assess the impact of PBM measures on mortality. The majority of the studies determined whether a restrictive transfusion strategy would be clinically non-inferior to a liberal strategy. Include a statement (and corresponding citation of one of two trials included in the meta-analysis (TRICCSIII/Carson et al/REALITY) to the discussion. 6) Include a sentence to the discussion section about the overall number of studies and patients included in this secondary analysis. 7) Page 16, line 30: delete “that was restricted to the 5 most common PBM interventions.” and cite two or three studies published in recent years elucidating the impact of one of the 5 investigated measures. 8) The analysis included studies published till 1st June 2019. This secondary analysis is not timely, as several (high ranking) studies have been published recently. Add a statement, that the search finished in June 2019 and that high ranking studies published after the search was finished were not included in this analysis. 9) Revise the sentence “Patient Blood Management moves this one step further, advocating the implementation of multiple interventions to prevent the use of blood, on the basis that this results in improved outcomes for patients or cost effectiveness” according to the most recent publication about the definition of PBM (Shander et al, DOI: 10.1213/ANE.0000000000005873) 10) The author state in their rebuttal letter, that the number of studies (n=4 including 761 patients) associated with the conclusion “Third we observed that trials where authors had declared links to professional PBM advocacy organisations reported statistically significant reductions in mortality, unlike other groups.” were included in the first paragraph of the discussion. I do not find any information regarding this comment. 11) The author state in their rebuttal letter, that the analysis is unable to measure the direct influence of PBM advocacy groups on trial conduct and reporting. For this reason, the authors used the ICMJE reporting standards against which individual trial reports were assessed. Whilst this cannot point to specifics, it provides an international consensus framework for assessing and reporting conflicts. Please add this information to the limitation section of the manuscript. 12) The authors used RR to measure effect size, thereby excluding studies with zero events from analysis. Please include a statement in the limitation section of the manuscript regarding omitting studies with zero events. How many studies were omitted from the analysis? Provide information in the method or result section. 13) The risk of bias assessment shown in eFigure2 is misleading as no distinction was made for the investigated PBM interventions. Here, risk of bias for allocation is judged with high risk more often in studies using cell saver compared to studies using TXA for example. Include a summary of the risk of bias assessment for the investigated single interventions/studies in the result section.
--	--

	14) Finally, include a statement about the role of reviewers and editor as a source for bias in the discussion section. 15) Thank you very much for updating the funding sources. One of the authors declared COIs in terms of a director of The Iron Clinic Ltd and director of Veincare London Ltd & Veincare WA also is the Vascular lead for 18-week wait Ltd. What does this mean? Are there any financial benefits for the author? In which way readers of the article may be better informed of this COI?
--	--

VERSION 2 – AUTHOR RESPONSE

Referee: 2

Comments to the Author

Thank you for answering my questions, and for the adaptations made. I consider that the revisions improved considerably the text.

Answer: Thank you for helping us improve the quality of our manuscript.

The authors performed an extensive analysis with several important results. To enhance the transparency and prevent any reporting bias I suggest to include to each discussed conclusion the corresponding result: which analysis was performed, the number of analysed studies and patients.

Answer: Thank you for this recommendation. The Main findings has now been updated to include the type of analysis performed, number of analyses studies and number of patients.

Thank you very much for clarifying the line of thoughts for the conclusion stated in the first paragraph on page 14 (page 14, lines 15-35). Please include the following statement in line 15/page 14: "These results clearly show that: 1. The evidence indicates that PBM interventions reduce transfusion. 2. Funnel plots and Egger's tests are highly suggestive of reporting bias. 3. Fill and trim demonstrated that the reporting bias was in favour of the treatment effects of PBM on reducing transfusion. We therefore interpret these results as showing clear links between reporting bias and the magnitude of the treatment effect on transfusion, one of our primary endpoints."

Answer: Thank you for this suggestion. This has now included in the discussion.

The authors found that trials in which authors had declared links to professional PBM advocacy organisations reported statistically significant reductions in mortality, unlike other groups. This conclusion is based on the analysis of 4 studies including 761 patients and RR 0.4 (95%CI 0.17-0.96); p<0.04). Please include this information to the corresponding section in the discussion and provide information whether the 4 studies analysed the same PBM intervention. As stated in your rebuttal letter, include also the following information: we acknowledge that the analysis is unable to measure the direct influence of PBM advocacy groups on trial conduct and reporting.

Answer: Thank you for the suggested clarification. The clarification "(4 studies with 761 patients)" has been added to the corresponding results section on page 12. Results are usually not included in the discussion section. Information on the PBM intervention analysed was included in the limitation section. The information "we acknowledge that the analysis is unable to measure the direct influence of PBM advocacy groups on trial conduct and reporting." was included in the limitation section.

Include the following statement from your rebuttal letter to the discussion section: "Our secondary outcomes did not have these qualities (eTable 2 in the Supplement) with heterogeneity in disease definitions, reported outcomes, and estimated treatment effects. The definition of adverse events in particular was very heterogeneous between studies, limiting assessment of this data. Overall, 8/102 secondary outcome analyses for important clinical outcomes stratified by type of conflict yielded a p value for treatment effect <0.05. Analyses of bleeding and transfusion outcomes generally favoured PBM, as per the findings of our primary analysis of red cell transfusion."

Answer: Thank you for this suggestion. The statement "Our secondary outcomes analyses demonstrated (eTable 2 in the Supplement) heterogeneity in disease definitions, reported outcomes, and estimated treatment effects. The definition of adverse events in particular was very heterogeneous between studies, limiting assessment of this data. Overall, 8/102 secondary outcome analyses for

important clinical outcomes stratified by type of conflict yielded a p value for treatment effect <0.05. Analyses of bleeding and transfusion outcomes generally favoured PBM, as per the findings of our primary analysis of red cell transfusion." Has been added at the end of the main discussion.

PBM is a multimodal concept consisting of several measures that in combination enhance patient safety. Studies included in the secondary analysis were not designed to assess the impact of PBM measures on mortality. The majority of the studies determined whether a restrictive transfusion strategy would be clinically non-inferior to a liberal strategy. Include a statement (and corresponding citation of one of two trials included in the meta-analysis (TRICCIII/Carson et al/REALITY) to the discussion.

Answer: We acknowledge that the majority of the studies were not designed to assess the impact of PBM on mortality. A statement to reflect this has been added to the limitations. We politely disagree that the majority of the studies were comparing liberal and restrictive transfusion strategies. From the 93 studies in 26766 patients, only 19 studies with 12866 patients compared liberal and restrictive transfusion strategies, while the majority of the studies comparing Tranexamic acid and controls (47 studies with 10621 patients). Despite the lower number of studies, the majority of the patients as suggested by the reviewer were present in the restrictive vs. liberal studies. This is detailed in the BJA manuscript (<https://doi.org/10.1016/j.bja.2020.04.087>).

Include a sentence to the discussion section about the overall number of studies and patients included in this secondary analysis.

Answer: This information has been included in the Clinical Importance section.

Page 16, line 30: delete "that was restricted to the 5 most common PBM interventions." and cite two or three studies published in recent years elucidating the impact of one of the 5 investigated measures.

Answer: Thank you for this suggestion. This has now been amended, and two recent studies have been referenced ([https://doi.org/10.1016/S0140-6736\(20\)31539-7](https://doi.org/10.1016/S0140-6736(20)31539-7) and <https://doi.org/10.1001/jama.2021.0135>)

The analysis included studies published till 1st June 2019. This secondary analysis is not timely, as several (high ranking) studies have been published recently. Add a statement, that the search finished in June 2019 and that high ranking studies published after the search was finished were not included in this analysis.

Answer: Thank you for raising this limitation. A statement has been added to the discussion section.

Revise the sentence "Patient Blood Management moves this one step further, advocating the implementation of multiple interventions to prevent the use of blood, on the basis that this results in improved outcomes for patients or cost effectiveness" according to the most recent publication about the definition of PBM (Shander et al, DOI: 10.1213/ANE.0000000000005873)

Answer: Thank you for this suggestion. This paragraph has now been amended to read: " Patient blood management has been recently defined as a patient-centred, systematic, evidence-based approach to improve patient outcomes by managing and preserving a patient's own blood, while promoting patient safety and empowerment.(21) Patient Blood Management moves this one step further Recent guidelines, advocating the implementation of multiple interventions to prevent the use of blood, on the basis that this results in improved outcomes for patients or cost effectiveness.(2)"

The author state in their rebuttal letter, that the number of studies (n=4 including 761 patients) associated with the conclusion "Third we observed that trials where authors had declared links to professional PBM advocacy organisations reported statistically significant reductions in mortality, unlike other groups." were included in the first paragraph of the discussion. I do not find any information regarding this comment.

Answer: This is now included as per the previous point made by the reviewer.

The author state in their rebuttal letter, that the analysis is unable to measure the direct influence of PBM advocacy groups on trial conduct and reporting. For this reason, the authors used the ICMJE reporting standards against which individual trial reports were assessed. Whilst this cannot point to specifics, it provides an international consensus framework for assessing and reporting conflicts. Please add this information to the limitation section of the manuscript.

Answer: Thank you for this suggestion. The paragraph now reads: “Journal adherence to declarations of conflicts improved after the introduction of ICMJE reporting standards, which provides an international consensus framework for assessing and reporting conflicts, however these standards were present only in a minority of trials”

The authors used RR to measure effect size, thereby excluding studies with zero events from analysis. Please include a statement in the limitation section of the manuscript regarding omitting studies with zero events. How many studies were omitted from the analysis? Provide information in the method or result section.

Answer: The reviewer is correct to highlight that zero events are excluded from Risk Ratio analyses. This is however widely accepted in meta-analyses and is part of Cochrane methodology. (https://handbook-5-1.cochrane.org/chapter_16/16_9_3_studies_with_no_events.htm#:~:text=The%20standard%20practice%20in%20meta,of%20the%20relative%20treatment%20effect.) In outcomes with low event rates such as mortality, 11 of 93 (11.8%) studies had 0 events in both treatment and control arms. On further review the risk difference analysis that includes 0 events showed no difference in the statistical significance of overall and subgroup effects for these outcomes.

The risk of bias assessment shown in eFigure2 is misleading as no distinction was made for the investigated PBM interventions. Here, risk of bias for allocation is judged with high risk more often in studies using cell saver compared to studies using TXA for example. Include a summary of the risk of bias assessment for the investigated single interventions/studies in the result section.

Answer: Thank you for this comment. The risk of bias assessment based on the type of PBM intervention is available in the published supplement of the primary analysis (<https://doi.org/10.1016/j.bja.2020.04.087>) and is readily available through open access. We do not believe that the duplication of this data will add further value to the result section or interpretation of the main findings.

Finally, include a statement about the role of reviewers and editor as a source for bias in the discussion section.

Answer: Thank you for this request. There are no apparent conflicts of interest for the editor or reviewers, while potential biases in publishing have been previously discussed and are a topic of ongoing debate (<https://doi.org/10.1007/s11098-017-0895-4>). This may be particularly relevant in open peer review. The following statement has been added to the Conflict of Interest section: “No conflicts of interest relevant to this manuscript were disclosed by the reviewers or editor. The authors are unable to assess the sources of bias associated with the reviewers or editor in the open peer review process.”

Thank you very much for updating the funding sources. One of the authors declared COIs in terms of a director of The Iron Clinic Ltd and director of Veincare London Ltd & Veincare WA also is the Vascular lead for 18-week wait Ltd. What does this mean? Are there any financial benefits for the author? In which way readers of the article may be better informed of this COI?

Answer: Thank you for highlighting this. On further clarification with Prof. T. Richards, we can confirm that none of these COIs have any direct relationship or influence on the manuscript presented. The types of conflicts for TR are further detailed in the ICMJE declaration form submitted. A statement to clarify this has been added to the COI statement section.

VERSION 3 – REVIEW

REVIEWER	Suma Choorapoikayil Forschungszentrum Europäische Aufklärung
REVIEW RETURNED	03-Jun-2022
GENERAL COMMENTS	Thank you for answering my questions, and for the adaptations made. Please revise the following sentence in the result section of the abstract: Trials with declared conflicts linked to professional PBM advocacy groups (5 studies; n=977 patients) reported statistically significant reductions in mortality RR 0.40 (0.17-0.92), unlike other

	groups. I have no further suggestions and I am looking forward to see the study published.
--	--

VERSION 3 – AUTHOR RESPONSE

Referee: 2

Thank you for answering my questions, and for the adaptations made.

Answer: *Thank you for helping us improve the quality of our manuscript.*

Please revise the following sentence in the result section of the abstract: Trials with declared conflicts linked to professional PBM advocacy groups (5 studies; n=977 patients) reported statistically significant reductions in mortality RR 0.40 (0.17-0.92), unlike other groups.

Answer: *Thank you for this recommendation. This has now been changed in the abstract for clarity.*

I have no further suggestions and I am looking forward to see the study published.

Answer: *Thank you for all your support in publishing our manuscript.*